

# Fire activity in the northern Arctic tundra now exceeds late Holocene levels, driven by increasing dryness and shrub expansion

Angelica Feurdean[1,2*], Randy Fullweber,[3] Andrei-Cosmin Diaconu[4], Graeme T. Swindles[5,6] Mariusz Gałka*[7]

[1]Institute of Physical Geography, Goethe University Altenhöferallee 1, 60438 Frankfurt am Main, Germany Frankfurt am Main, Germany

[2]STAR-UBB Institute, Babeş-Bolyai University, Kogălniceanu 1, 400084, Cluj-Napoca, Romania

[3]Toolik Field Station, Institute of Arctic Biology, University of Alaska Fairbanks, United States of America

[4]Babes-Bolyai University, Department of Geology, Kogălniceanu 1, 400084, Cluj-Napoca, Romania

[5]Geography, School of Natural and Built Environment, Queen's University Belfast, Belfast, UK

[6]Ottawa-Carleton Geoscience Centre and Department of Earth Sciences, Carleton University, Ottawa, Canada

[7]University of Lodz, Faculty of Biology and Environmental Protection, Department of Biogeography, Paleoecology and Nature Conservation, Banacha 1/3, 90-237 Lodz, Poland

*Correspondence to*: Angelica Feurdean (feurdean@em.uni-frankfurt.de) and Mariusz Gałka (mariusz.galka@biol.uni.lodz.pl)




## Abstract

Tundra ecosystems are characterized by small, rare and infrequent fires due to cold, often waterlogged conditions, and limited biomass. However, ongoing climate warming and drying in northern soils and peatlands contribute to increasingly frequent and extensive wildfires. To place recent fire regimes in the context of long-term variability and to better understand interactions between moisture, vegetation, and fire, we reconstructed wildfire history over the past 3000 years using a network of charcoal records in combination with vegetation and hydrological datasets and satellite-derived fire datasets from northern Arctic

Alaska. Composite charcoal records show minimal fire activity from ~1000 BCE to 500 CE, followed by a slight increase from 500 CE onwards. This long-term pattern shifted abruptly around 1880 CE, when fire activity exceeded any levels observed in the preceding three millennia. Individual charcoal records show a more heterogeneous fire pattern before 1880 CE and a more homogeneous one thereafter. Our findings suggest that deepening of water tables and peatland drying associated with permafrost thaw have facilitated woody encroachment, especially by more flammable Ericaceous shrubs. These vegetation

changes have increased fuel availability and flammability, ultimately driving the recent surge in wildfire activity. This study highlights the importance of moisture–vegetation–fire feedback in shaping tundra fire regimes and the vulnerability of Arctic ecosystems to fire. This is particularly evident in areas experiencing pronounced drying and the expansion of flammable shrub taxa. We also found that the charcoal source area of our tundra fire encompasses broader landscapes over tens of kilometre.

## 1 Introduction

Short-term records of fire activity derived from satellite-based observations show that many northern tundra regions have experienced an unprecedented increase in the number, size, intensity, duration, and length of the fire season in recent decades (Descals et al., 2020; Scholten et al., 2021; 2024). The Arctic tundra biome occupies some of Earth's coldest regions, with low aboveground biomass productivity. However, Arctic regions store approximately half of the global belowground soil organic

carbon in peatlands (Loisel et al., 2021). Cold and waterlogged conditions in Arctic soils and peatlands often constrain fires to small, rare, and infrequent events and limit burning depth and carbon loss (Archibald et al., 2013; Turetski et al., 2015; Whitman et al., 2018; 2019; Saedi et al., 2024). Recent rising temperatures, increased lightning frequency, changes in peatland hydrology, and greater shrub biomass contributed to deviation from the low fire pattern (Scholten et al., 2024). Well-hydrated peatlands composed of moss and sedges can be resistant to droughts and exhibit only moderate surface fire damage compared

to peatlands dominated by shrub and tree communities (Magnan et al., 2012; Kettridge et al., 2014; Whitman et al., 2018). Deep burns can smoulder over winter in hydrologically disturbed peatlands and re-ignite during spring (Scholten et al., 2021).

      While satellite data offer high temporal and spatial resolution of many aspects of the fire regime, they are limited to the past few decades. Therefore, satellite records alone cannot reveal how recent aspects of fire regimes compare to longer-term variability. Charcoal analysis from sedimentary records provides critical insights into the long-term dynamics of fire,

offering essential spatio-temporal context for assessing recent changes. Combined with reconstructions of vegetation communities and hydrological conditions, it allows for the investigation of past tundra fire regimes under various climatic conditions and vegetation compositions (Higuera et al., 2018; Vachula, 2020; Hoecker et al., 2020; Sim et al., 2023). Therefore,



approaches with a broad temporal and spatial scope allow an evaluation of feedback between fire and drivers and form a foundation for understanding how tundra fire regimes may respond under future climate and vegetation development scenarios.

Although some palaeoecological records of tundra fires in Arctic Alaska exist (Hu et al., 2010; Chipman et al., 2015 Chipman and Hu, 2017; Gałka et al., 2018; Higuera et al., 2008; Vachula, 2020; Hoecker et al., 2020), combined records documenting fire, plant species composition, and hydrological variability from peatland are rare, limiting our understanding of the interaction between fires, vegetation, and hydrological conditions. Another limitation in determining past tundra fires is understanding charcoal source areas and the amount of charcoal incorporated in the fossil records during tundra fire (Vachula

et al., 2020).

     To place modern fire regimes in the context of long-term variability, we reconstructed 3,000 years of wildfire history using seven new and two previously published macrocharcoal records from a tundra region north of the Brooks Range in Arctic Alaska. We combined satellite and the charcoal records primarily from the dwarf-shrub and tussock sedges-moss- dwarf shrubs zone with data on local (plant macrofossils) and regional (pollen) vegetation and hydrological changes (testate amoebae–based

depth-to-water reconstructions) to investigate how interactions between peatland hydrology and vegetation affect fuel availability, dryness, and flammability. We hypothesize that the recent increase in fire frequency in northern Alaska is linked to the lowering of the permafrost layer and deepening of the water level and, consequently, the expansion of shrubs on dry peatlands. We also aim to refine the understanding of charcoal source areas in tundra fire reconstructions by comparing our charcoal records with satellite-derived data on fire size and frequency.


## 2. Geographical location and sites selection

The study area is located within the continuous permafrost zone of northern Alaska, on the northern side of the Brooks Range Mountains (Fig. 1). The landscape features a mosaic of peatland types, small glacial lakes, kames, and moraines (Hamilton, 1986). The region experiences a continental Arctic climate, with a mean annual temperature of approximately 0 °C. Mean

seasonal temperatures range from –22.5 °C in winter to +11.2 °C in summer. Annual precipitation averages 25 mm, falling predominantly as snow (Environmental Data Center Team, 2017). The number of snow-free days is approximately 110, although it can snow briefly during summer (Cox et al., 2017; https://akclimate.org/data/annual-reports). Vegetation is dominated by sedges, brown mosses, *Sphagnum*, and shrub species with composition varying by site (Walker et al., 1994, 2005; Table 1; Appendix A).

The study sites were distributed along a 150 km transect parallel to the Dalton Highway, extending from the northern foothills of the Brooks Range to the Arctic Ocean (Fig. 1). In the summer of 2015, we collected nine cores from six peatlands and one dry up hollow. Cores were retrieved using a long-bladed shovel until the mineral substrate was reached. All cores were placed in PVC tubes, transported, and deposited in a cold room at the University of Łódź, Poland (Gałka et al., 2018). Details on geographic coordinates, elevation, core lengths, and dominant vegetation at the coring points are provided in Table

95 1.



## 3 Material and methods

### 3.1 Chronology

Chronologies were established using radiocarbon dating via Accelerator Mass Spectrometry (AMS), applied to a combination of plant macrofossils and bulk peat samples, supplemented by $^{210}$Pb dating for recent sediments. Age–depth models for Toolik I and II (TFS I and II) are published in Gałka et al. (2018), those for Ga I and Ga II are provided in Gałka et al. (2023), and models for NBb and Erin are presented in Gałka et al. (2025). Chronological data for the SG, RH, and DL sites are presented in Appendix B.

### 3.2 Macroscopic charcoal inferred fire history

We reconstructed changes in local-scale fire activity based on macroscopic charcoal analysis of 1–2 cm³ sample volume taken at 1 cm contiguous intervals across all sites. Sample preparation involved overnight bleaching and wet sieving through a 160 μm mesh. Charcoal identification and categorization of morphotypes (later conducted for 6 of the eight records) followed the protocols of Feurdean et al. (2017) and Feurdean (2021). Charcoal morphologies were then grouped into non-woody 110 (graminoids and forbs) and woody to determine the dominant fuel type. We also separate the charcoal fractions (5 out of 8 records) into two size classes, 150–500 μm and >500 μm, to distinguish more local from more regional fires (Adolf et al., 2018). Charcoal influx (particles cm² yr) was calculated by dividing the particle concentration (particles cm³) by the deposition time derived from the age-depth models (yr cm). Additionally, we recorded large macrocharcoal fragments in samples used for plant macrofossil analysis. For this, 12 cm³ contiguous sediment samples were washed and sieved using a 200 μm mesh 115 without applying bleaching following Gałka et al. (2018).

### 3.3 Composite records of fire, vegetation and hydrological changes

To assess fire history across the study area, we constructed a composite record of charcoal accumulation rates (CHAR) by combining all individual records. We generate a composite CHAR record following the protocol of Blarquez et al. (2014), 120 which included a MinMax re-scaling, a Box-Cox transformation for homogenizing variance across records, and a Z-score standardization using a baseline period spanning from 3 ka to the present, smooth with a LOWESS smoother set to a window width of 200 years. Confidence intervals were estimated from the distribution of 1,000 bootstrapped replicates. We created a composite record of shrubs by pooling available pollen and plant macrofossil records at TFS I, TFS II, Ga I, Ga II, NBb, and Erin to determine the local and regional changes in woody vegetation. In the case of plant macrofossils, the root abundance is 125 represented as a percentage, whereas above-ground remains (seeds, leaves, fruits) are represented as counts. We used available testate-amoeba-based water table reconstruction (DTW) at TFSI, TFSII, GaI, GaII, and Erin to create a composite water table depth record. We apply a LOWESS smoothing set of 200 years for all composite records; however, this was done on Z-score transformation in the water table composite record case (Swindel et al., 2019).

### 3.4 Fire identification from satellite images



To determine recent trends in fire activity, evaluate our understanding of charcoal source areas, and the amount of charcoal produced in tundra fire, we downloaded two datasets available from the Alaska Interagency Coordination Center (AICC) website: Alaska Fire History Location Points (fire points) and Alaska Fire History Perimeter Polygons (fire polygons) (Miller et al., 2023). Spatial analysis was conducted using ArcGIS v. 3.4.3 with the coordinate system set to the Alaska Albers Equal Area Conic (EPSG: 3338) projection. We identified those fire points (n=35) and fire polygons (n=9) that fell within 2, 25, and 70km from the Dalton Highway, north of the Brooks Range (Fig. 1). We calculated the distance from each fire point and fire polygon from the Dalton Highway as well as the distance from nearest five fire points to each of the fossil record. For the fire point data, an estimate of the area burned is provided in hectares. For the polygon data, the size of the fire is reported as area in hectares, determined by the ArcGIS' Calculate Geometry Attributes tool (S1).

## 4. Results

### 4.1 Charcoal based biomass burning

Of the nine macrocharcoal records analysed, seven span the last one to two millennia, while only two extend as far back as 3000 BCE, albeit at low temporal resolution (Fig. 2). Charcoal accumulation rates (CHAR) are consistently low across all sites, ranging from 0 to 1.07 particles $cm^{-2}$ $yr^{-1}$ (mean= 0.04 ± 0.11). The highest mean CHAR values were observed at SG (0.05 particles $cm^2$ $yr^1$) and NBb (0.09 particles $cm^2$ $yr^1$).

The individual and composite CHAR records show consistently low values between 3000 and 1880 CE (negative Z-scores). However, a slight increase in composite CHAR was visible from 500 CE onwards, but mostly around 500, 700, 1200-1700 CE, though Z-scores remained negative. A sharp increase in composite CHAR started around 1880 CE, reaching a maximum between 1890 and 1970 CE (positive Z-scores), an increase depicted by all individual records. There was a slight CHAR decline thereafter till the present times, with temporarily higher values around 1986 and 2004 CE (Fig. 3).

The charcoal morphological record indicates the almost equal representation of herbaceous (e.g., grass leaves and stems of grasses and forbs) and woody charcoal morphologies, with higher woody influx towards recent times (Fig. 2). Small-size charcoal particles (150–500 μm) dominate over the larger fraction (>500 μm) (Appendix C). No large macro charcoal fragments (>1 mm) were detected in plant macrofossil records.

### 4.2 Comparison of historical fire data and charcoal

Satellite imagery record of fire points within a 70 km radius along the Dalton Highway shows that 36 fires occurred between 1969 and 2023. Except for an extremely large fire of 103.896 ha occurring in 2007, most recorded fires were of small size (mean = 46 ha; median =2 ha). Fire polygon data indicate nine fires with a mean size of 11.825 ha (median 115 ha; Fig. 4; S1). There were 10 recorded fires (0.5 per year) between 1969 and 1990, followed by a fire-free period from 1990 to 2001 (Fig. 4ab). Fire activity increased markedly between 2001 and 2017, with 23 recorded fires (1.5 per year), before declining again, with only one fire observed in 2023. Satellite imagery also revealed that the five closest historical fire points were located within a mean distance of 12, 20, 29, 34 and 41 km of the charcoal sampling sites (Figs. 1, 4c; S1).





On a temporal scale, we found that the nearest fires to the sampling points (range 3-29 km; mean 12 km) occurred in years 1971, 1979, 1990, and 2007, and the nearest sampling points to these fires were Toolik II (3 km), Dry Lake (6.5 km), and RH (7 km). The second closest fires from the charcoal sampling (range 15-31 km; mean 20 km) occurred in years 1969, 1970, 1971, 1979, 1983, 1990, 2004, 2007, 2017 (Fig. 4 c; S1). Given the inherent chronological uncertainties in sedimentary records and the very low amount of charcoal found in our records, the slight increases in charcoal influx centred around the

years 1960s,1970s, 2000s, 2010s, could be associated with documented fire events occurring in seasons 1969–1983 and 2007-2014 (Figs 1, 4b; S1). Notably, the charcoal-based signal indicates peak burning during 1969–1990, while the satellite data during 2001–2017 CE (Fig. 4a).

## 5. Discussion and conclusion

### 5.1 Charcoal source areas in tundra fire reconstructions

 Our analysis of charcoal size classes indicates that most charcoal particles fall within the 150–500 μm range, with only a small proportion exceeding 500 μm and no large macrocharcoal fragments (>1 mm) identified (Fig. 2; Appendix C). The dominance of smaller charcoal particles across our sites and the findings of nearest fires occurring within 3 to 29 km of the charcoal sampling points suggest that the charcoal source areas of our tundra fires encompass broader landscapes (Figs. 1,4c). Although

typical source areas for macrocharcoal are estimated at 2 km, regional source areas extending to 20 km and beyond (i.e, up to 100 km) have also been documented. This is particularly evident for smaller particles, those that are lighter and more aerodynamic (herbaceous) or under the specific wind and fire behaviour conditions (Higuera et al., 2005; 2007; Adolf et al., 2018; Vachula et al., 2020; Vachula and Rehn, 2023). We also found that the most recent large fires (period 2001-2015) have a weaker charcoal signal or are not captured by the charcoal records (Fig. 2, 4c). Graminoids have significantly lower charred-

mass retention per unit biomass than forbs and shrubs, and high fire temperatures can result in near-complete biomass consumption, producing mostly ash and limiting charcoal formation and preservation (Pereboom et al. 2020; Feurdean et al., 2021).

### 5.2 Pattern in Arctic Alaska peatland burning

Our satellite-based fire database from the northern slopes of the Brooks Range along the Dalton Highway agrees with previous findings that many tundra ecoregions of Alaska have experienced a marked increase in fire frequency, size, and intensity over the past five decades (Higuera et al., 2008; Vachula, 2020; Scholten et al., 2024; Miller et al., 2024). Specifically, our data show that lightning-ignited fires occurred at an average rate of 0.5 per year between 1969 and 1990, followed by a fire-free interval from 1990 to 2001 CE. Fire activity surged again from 2001 to 2017 CE, averaging one fire every 1.5 years, before

declining in the most recent years (Fig. 4ab). Historical evidence reaching back to the 18th century further document more tundra fires in the northern Brooks Range between 1880 and 1920 (Miller et al., 2024).

   Our individual and composite charcoal records spanning the past three millennia reveal little fires in the Arctic tundra from ~1000 BCE to 500 CE (Figs. 2, 3). Although there was a slight increase in fire activity from 500 CE onwards, this overall



low burning level was abruptly interrupted only around 1880 CE when charcoal evidence points to unparalleled fire increase.
The sharp rise in fire activity in recent decades compared to the rare and long fire-free interval during the late Holocene reinforces findings from other tundra ecoregions in Arctic Alaska (Higuera et al., 2008, 2011; Chipman et al., 2015; Hu et al., 2015; Vachula, 2020; Hoecker et al., 2020) highlighting that the more recent fire activity in these cold with low aboveground biomass is unprecedented.

### 5.3. Drivers of peatland burning

Recent increases in temperature and lightning frequency are often cited as key drivers of enhanced fire activity across Arctic tundra regions (Hu et al., 2010, 2015; Hoecker et al., 2020; Scholten et al., 2024; Sim et. al., 2023). Our independent reconstruction of peatland hydrology reveals that the lowest water table levels (positive Z score), indicating the driest conditions of the late Holocene, occurred during the past few decades, coinciding with the most intense fire period (Fig. 3). This recent deepening of water table was linked to warming-induced permafrost thaw (Taylor et al., 2019; Gałka et al., 2023; Cleary et al., 2024). Smaller-amplitude deepening in the water table between ~1250–1600 CE and around 250 CE were also accompanied by moderate increases in biomass burning (Fig. 3). However, the deepening of water tables in the composite record visible around 250 CE is driven primarily by deeper water levels at the TFS II site and only mirrored by the increased CHAR at this site, not the composite charcoal record (Figs. 2, 3). Lower temperatures and reduced precipitation, particularly during summer months of the Little Ice Age (~1250–1600 CE), may have contributed to peatland drying and water table deepening (Taylor et al., 2019; Zhang et al., 2022). These patterns suggest a persistent relationship between low peatland moisture and increased fire activity. Drier conditions promote the desiccation of surface fuels, increasing the potential for ignition and sustained burning (Katrigge et al., 2020; Feurdean et al., 2022).

Our multiproxy palaeorecords provide a unique opportunity to investigate local feedback between moisture conditions, fuel type and availability and fire particularly in areas prone to pronounced drying, such as hummocks. Plant macrofossil data from the same sites show the highest abundance and diversity of shrub communities, including *Betula nana*, undifferentiated Ericaceae, *Ledum*, *Andromeda*, *Vaccinium*, and *Salix*, after 1850 CE, with a notable increase after 1950 CE (Fig. 3). The pollen record, which reflects landscape-scale vegetation composition, shows a pattern of shrub expansion around 500–700 CE, 1200–1600 CE, and after 1900 CE, mirroring peaks in charcoal values (Fig. 3). This visual correlation between increased fire activity, reduced surface moisture, and expanded shrub cover suggests that woody biomass may have enhanced fuel quantity and flammability, thereby promoting fire. Although this relationship could not be confirmed statistically (not shown), our results nonetheless hint to the role for moisture–vegetation feedbacks in shaping fire regimes in Arctic tundra ecosystems. Additionally, a close inspection of fire occurrence and vegetation composition derived from vegetation maps shows that most recent fires in the study area occurred in tussock sedge–dwarf shrub and dwarf shrubs (Appendix A). Pollen and macrofossil evidence further indicate that *Alnus* and *Salix* were more common before 1850 CE. In contrast, Ericaceous shrubs and, to some extent, *Betula nana* have become more dominant in recent decades (Gałka et al., 2018, 2023, 2025). *Salix* and *Alnus* can grow on dry mineral soil, whereas Ericaceous shrubs usually grow on peaty soils. Ericaceous shrubs are more

flammable, while *Betula* species tend to resprout vigorously following fire (de Groot et al., 1999, 2013). A strong link between the expansion of resprouting taxa such as *Betula glandulosa* and elevated fire activity has been documented in north-central

Alaska during the Lateglacial and early Holocene (Higuera et al., 2008). We conclude that the development of more flammable or resprouting shrub species in peatlands suggests that future shrub-dominated tundra landscapes may become even more fire-prone than they were in the past.

**Appendices**:

**Appendix A.** The dominant vegetation type in the study region.

**Appendix B.** Chronology of unpublished sites.

**Appendix C.** Charcoal influx of particles larger then > 500 μm.

**Data availability:** The data will be deposited into Neotoma database

**Supplement S1** Satellite based information of fire names, coordinates, size, and distance from the Dalton Highway and charcoal sampling sites.

**Author contribution**: AF and MG conceive study. AF, ACD, RF, GS, MG, performed the analysis. AF prepared the manuscript with contributions from MG. All authors contributed to review and editing.

**Competing interests**: The authors declare that they have no conflict of interest.

**Acknowledgements**: We thank Liam Taylor for performing the testate amoeba analysis at TFS I and II.

**Financial support**: The research has received support from National Science Centre (Poland) grant No UMO-
2019/35/B/ST10/00367 (PI: Mariusz Gałka). AF acknowledges support Deutsche Forschungsgemeinschaft (grant no FE_1096/9*)* during writing stage.






**Appendix Fig. A1** The dominant vegetation type in the study region which depicts the main vegetation types where historical fires occurred. For details, see Fig. 1.

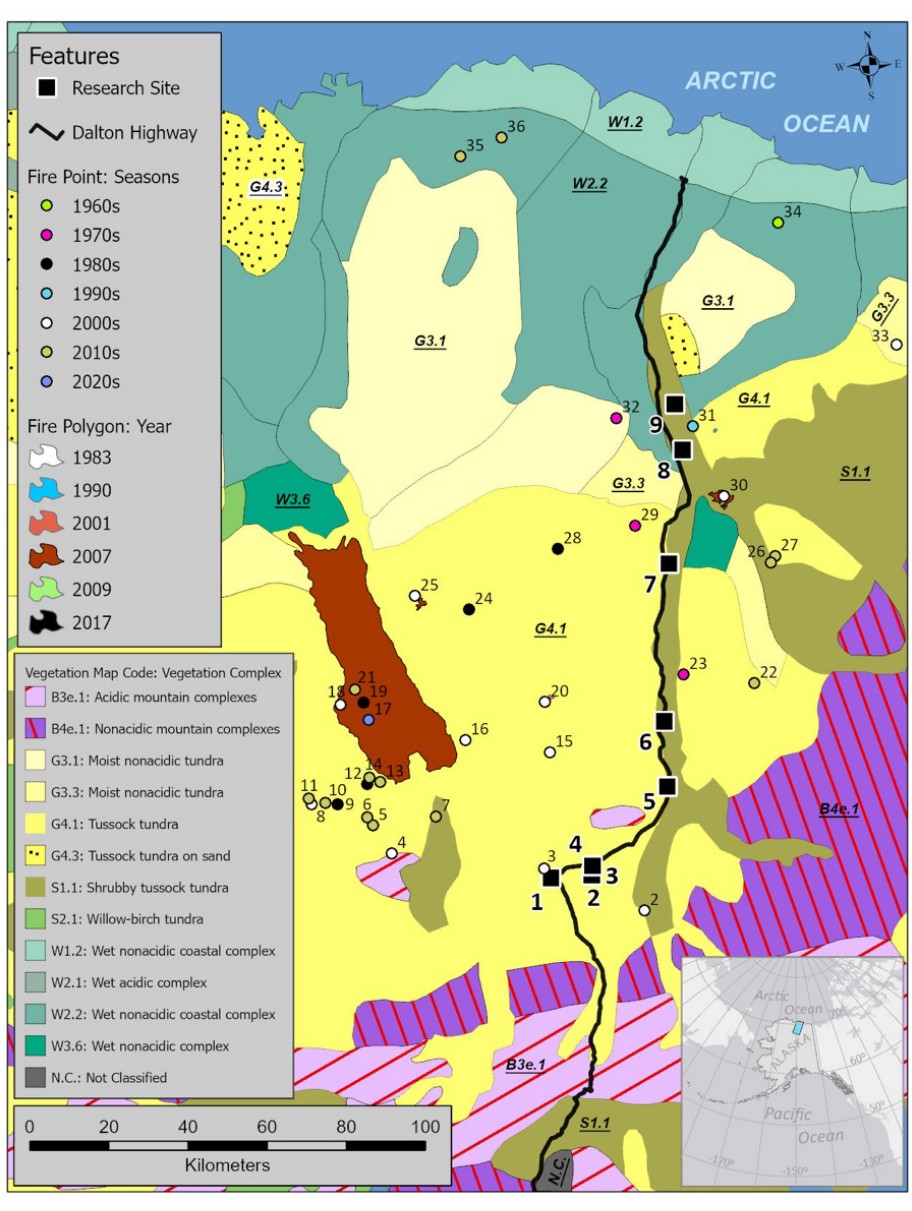




**Appendix B. Chronology**

AMS radiocarbon measurements of the unpublished sites SG, RH, and DL are presented in Table B1. The calibration of the radiocarbon dates and the construction of the age depth models were performed with OxCal 4.4.4 software (Bronk Ramsey, 2009) and the IntCal20 (Reimer et al., 2020) and post-bomb NH1 (Hua et al., 2022) calibration curves. The Bayesian age-

depth models were calculated by applying the *P_Sequence* function (parameters = 1 cm-1, $log10(k/k0) = 1$ and *interpolation* = 0.5 cm). BCE before Common Era. CE Common Era.

Table B1 AMS $^{14}$C measurements of at SG, RH and DL

| Depth (cm) | Material | Nr. Lab. | C14 date |
|---|---|---|---|
| SG 6.25 | *Scorpidium scorpioides* stems with leaves | Poz-151142 | 120.77 ± 0.35 pMC |
| SG 16.75 | *Carex* fruits and brown moss stems | Poz-151139 | 305 ± 30 BP |
| SG 26.75 | *Carex* fruits , *Betula nana* fruits, *Scorpidium scorpioides* leaves | Poz-151140 | 570 ± 30 BP |
| SG 36.75 | *Carex* fruits, *Scorpidium scorpioides* leaves | Poz-151141 | 1010 ± 40 BP |
| SG 48-49 | *Carex* fruits and stem | Poz-108707 | 1680 ± 35 BP |
| RH 8.75 | *Scorpidium* stems, *Betula nana* leaves | Poz-135148 | 122.89 ± 0.32 pMC |
| RH 17.25 | *Carex* fruits, *Carex* basal stems | Poz-135149 | 655 ± 30 BP |
| RH 23.25 | *Carex* fruits, brown moss leaves | Poz-135192 | 1460 ± 30 BP |
| RH 28.25 | wood pieces, brown moss stem | Poz-135193 | 1705 ± 30 BP |
| RH 35-36 | wood pieces | Poz-109880 | 5000 ± 40 BP |
| DL 5 | Brown moss | DeA-37949 | Modern; post 1950 |
| DL 25 | Brown moss and sedge | DeA-37950 | 1691 ± 21 BP |
|  |  |  |  |

**Fig. B1** Age depth model at SG (Fig. B1a), RH (Fig. B1b) and DL (Fig. B1c). The Y axis represents the depth (cm); the code

and position of each radiocarbon measurement are shown inside the plot. The light and dark blue shaded areas represent uncertainty ranges in the age-depth model. The black histogram shows the possible age of each measured sample.




Fig. B1a



Modelled date (BCE/CE)




Fig. B1b

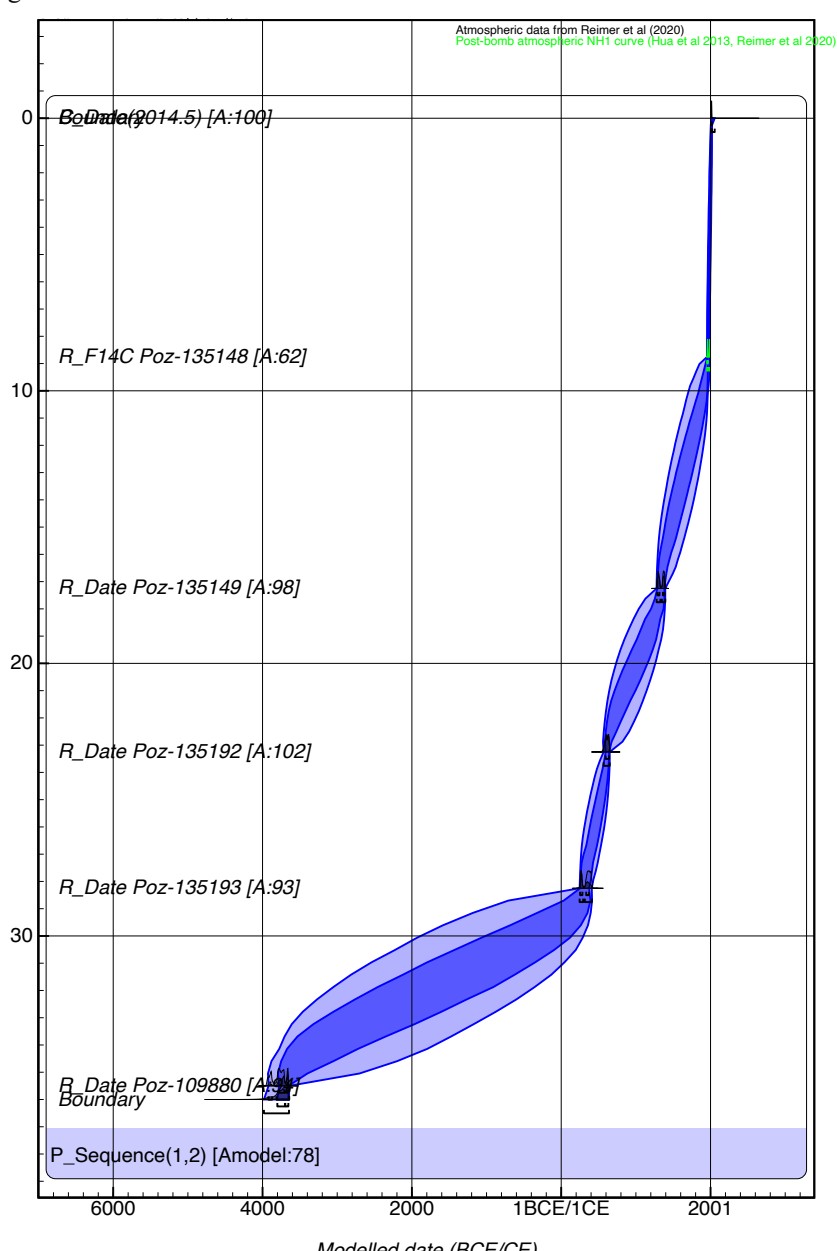




Fig. B1c

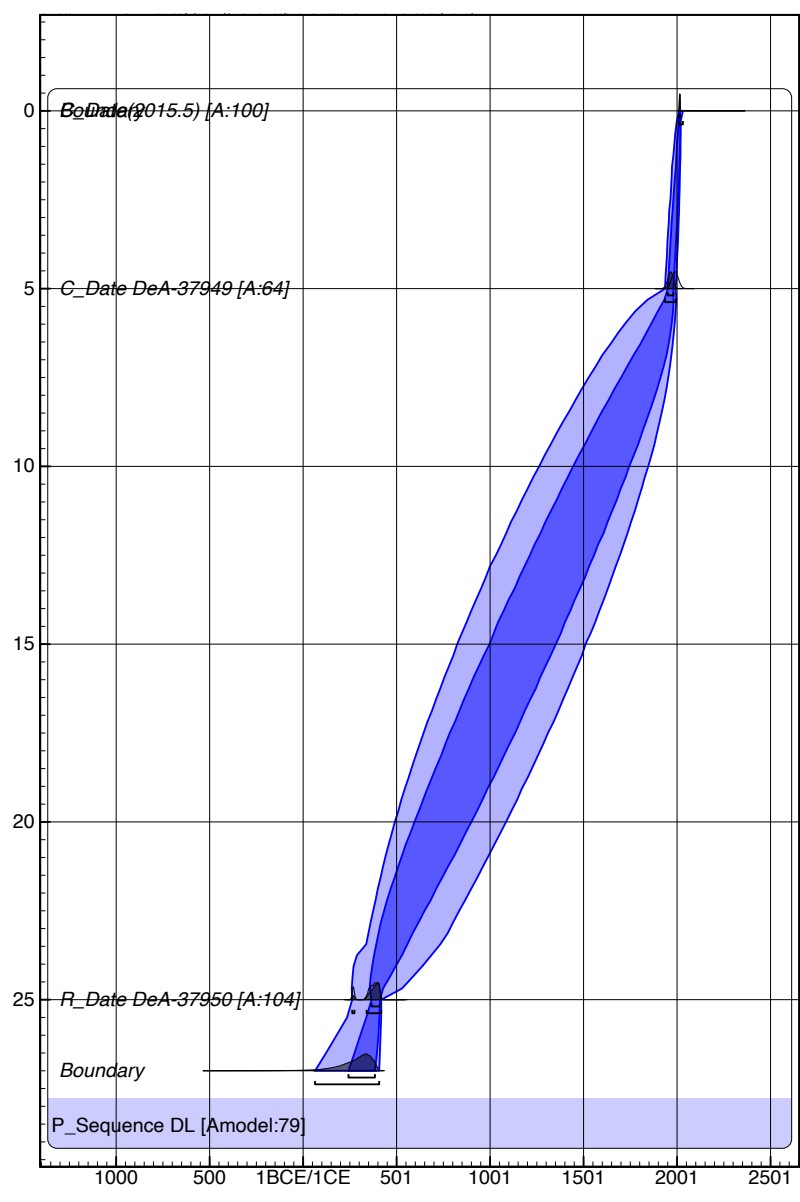





**Appendix Fig. C1** Charcoal influx (#/cm$^{-2}$ yr$^{-1}$) of particles larger then > 500 μm in the sequences with available counts size.

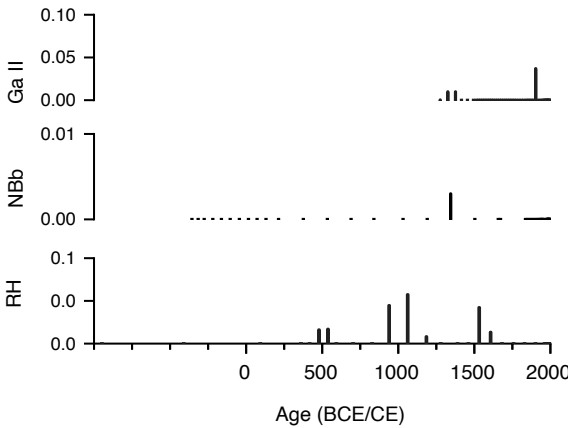

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



**Figure legends and embedded figures**

**Figure 1.** Location of the study area in northern Alaska. Charcoal sampling sites numbered 1 to 9 are situated along the Dalton Highway, extending from the northern slope of the Brooks Range Mountains to the Arctic Ocean. Site names and coordinates are listed in Table 1. Fires are represented as polygons (large fires) and points (small fires). The colour coding of each fire
polygon and point indicates the decade in which the fire occurred, while the numbers refer to their specific locations (for fire names, coordinates, size, and distance from the Dalton Highway and sampling sites see S1).

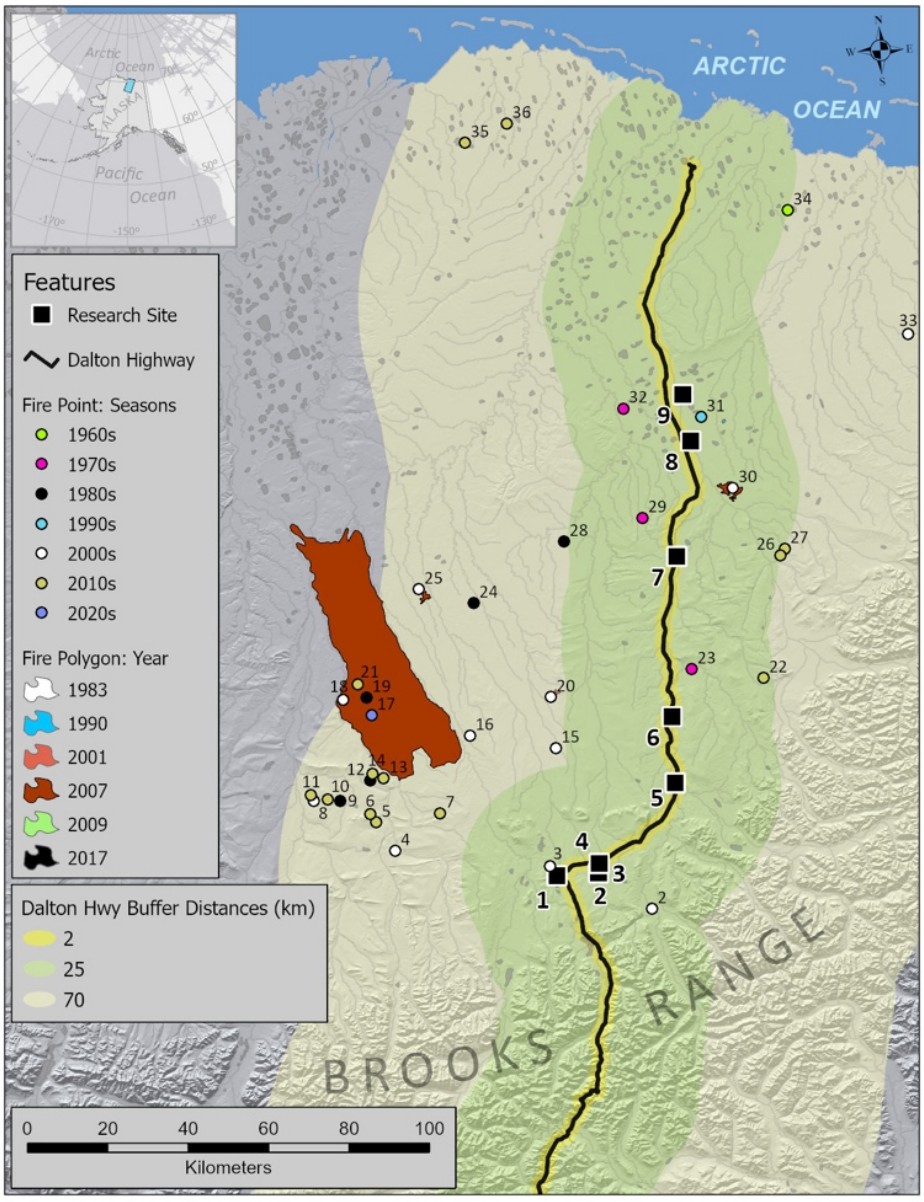





**Figure 2.** Total charcoal influx (black bars) and woody charcoal morphologies (green bars) #/cm²/yr) at nine sites along the Dalton Highway (for each site's exact location, see Fig. 1). The total charcoal influx represents the unidentified charcoal. At sites with identified woody morphologies, the difference between the woody morphology (green) and the total charcoal (black) represents the morphologies formed by herbs (grass leaves, stems of forbs, and grasses) and broadleaves. Orange rectangle highlights period of marked increase in total charcoal influx. BCE=before Common Era; CE= common Era. On the right hand are trends in charcoal influx over the last two centuries.

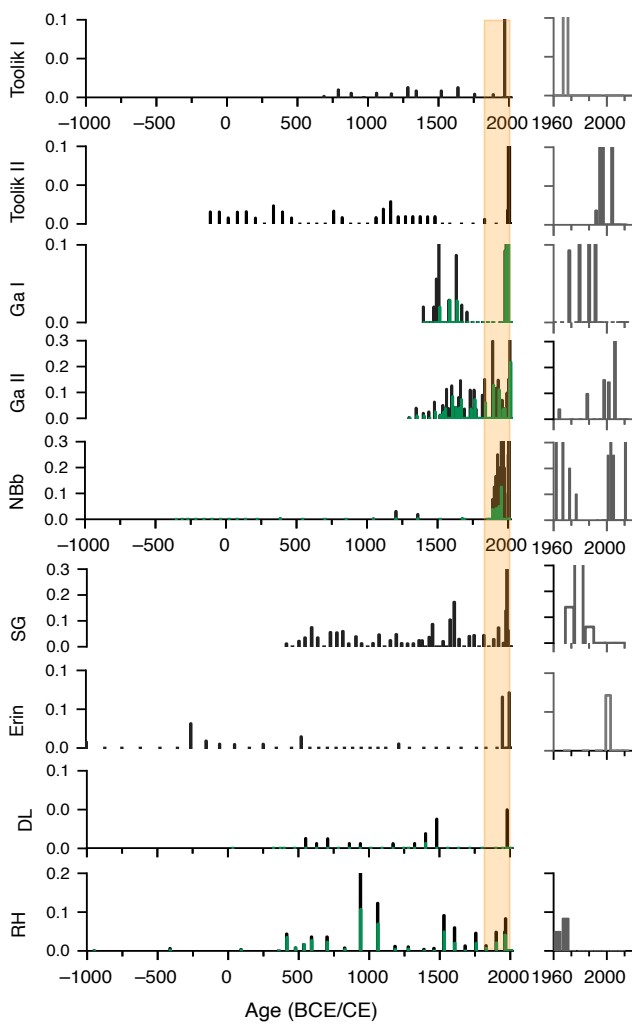



**Figures 3** a) Composite record of biomass burning (n=9) based on Z-score charcoal influx where positive/negative Z-score
values represent greater-than-mean/lower-than-mean charcoal influx over the base period. b) Composite record (n=5) of the
abundance of shrubs determined from pollen and plant macrofossils, where pollen and root shrub biomass is represented as %,
and above-ground remains (seeds, leaves, fruits) as counts (c). Composite record of peatland hydrology (n=5) from testate
amoeba, where positive/negative Z-score values represent higher (shallow) /lower (deep) than mean water level. Grey curves
represent confidence intervals. The inserts represent trends over the last two centuries.

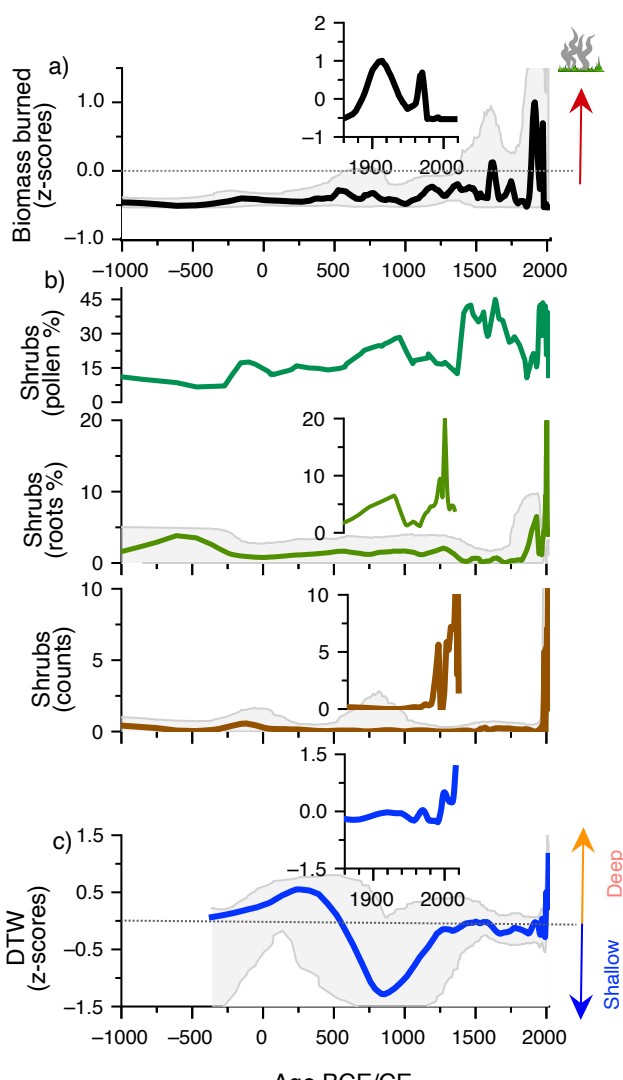



450 **Figure 4.** Comparison of fire frequency and size (ha) from satellite imagery (black bars) with composite charcoal influx (scattered lines; #/cm²/yr): a) fire size represented as fire points; b) fire size represented as fire polygons. *Note: The full extent of large fires is not shown to improve visibility of trends in smaller fire events. Specifically, the following fires are not shown in full: a) 2001 (341 ha) and 2007 (659 ha); b) the 2007 (103.896 ha). c) Distance to nearest five fires from each of the sites: I=denote the nearest and V= the furthest from the each of the sites.

455

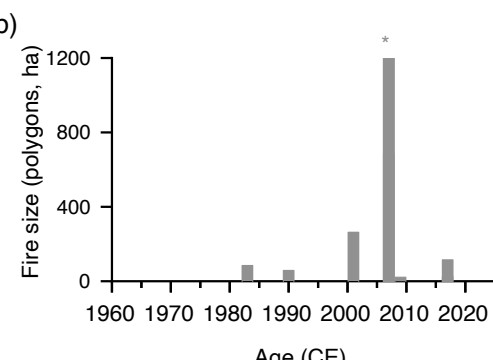

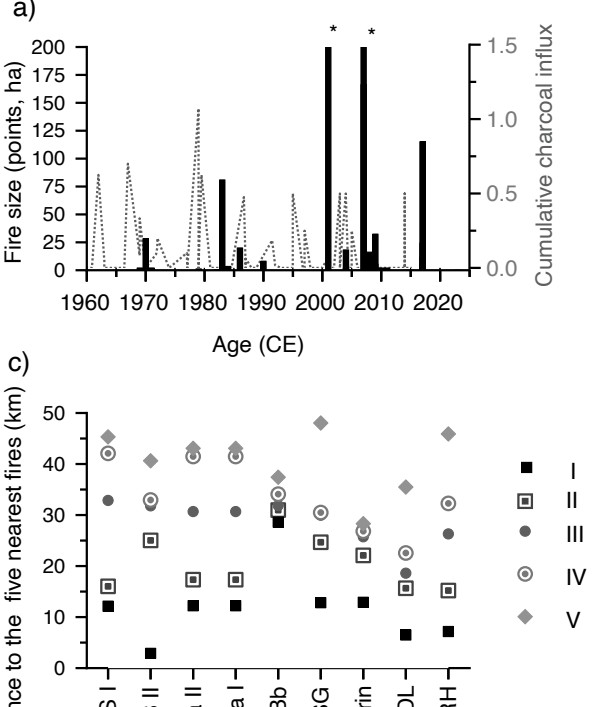

460




**Table 1** Information on geographic coordinates, elevation, core lengths (in cm and age) and the dominant vegetation at the coring points.

| Core | Co-ordinates | Elevation (m) | Core length (cm) | Age range (CE/BCE) | Dominant surface vegetation |
|---|---|---|---|---|---|
| TFS1 | 68.6239, 149.3458 | 715 | 45 | 2014-688 | *Sphagnum fuscum, S. capillifolium, Andromenda polifolia, Betula nana* |
| TFS2 | 68.6239, 149.5978 | 750 | 50 | 2010-118 BCE | *S. capillifolium, A. polifolia, Betula nana* |
| Gal I | 68.6453, 149.3375 | 876 | 46 | 2015-1400 | *Paludellla squarrosa, Carex* spp. *Andromenda polifolia, Betula nana,* |
| Gal II | 68.6453, 149.3375 | 876 | 50 | 2015-1300 | *Dryas octopetala, Salix* spp., *Rubus chamaemorus, Sphagnum teres, S. warnstorfii* |
| NBb | 68.8136, 148.8406 | 461 | 50 | 2015-300 BCE | *Scorpidium scorpioides, S. cossonii, Carex* spp., *Sphagnum subfulvum, Betula nana, Dryas octopetala, Salix* spp., *Rubus chamaemorus* |
| SG | 68.9647, 148.0841 | 159 | 50 | 2010-400 | *Carex* spp., *Tomentypnum nitens, Andromeda polifolia, Arctostaphyllos, Dryas octopetala, Salix* |
| Erin | 69.3233, 148.7208 | 230 | 44 | 2015-1125 BCE | *Carex* spp., *Tomentypnum nitens, Scorpidium cossonii, Betula nana, Dryas octopetala, Salix Rubus chamaemorus* spp. |
| DL | 69.5811, 148.5765 | 159 | 49 | 2008-3000 BCE | *Carex* spp., *Scorpidium scorpioides, Bryum pseudotriquetrum* |
| RH | 69.6875, 148.6017 | 156 | 36 | 2011-2600 BCE | *Carex* spp., *Scorpidium revolvens, Dicranum* sp., *Ledum groelandicum, Salix sp.* |