# Peer review of "Fire activity in the northern Arctic tundra now exceeds late Holocene levels, driven by increasing dryness and shrub expansion"

_EGUsphere, 2025_

## Author Response (AR1)

**30 August 2025, Frankfurt am Main**

We thank the reviewers and the editor for providing encouraging feedback and valuable comments on the manuscript, which helped enhance its accuracy and readability. We agree that some of the methods and interpretations, as well as parts of the discussions, were brief. In the revised manuscript, we expanded these parts based on these suggestions. Please find below our point-by-point response to all comments and suggestions raised by the reviewers and the editor. We marked our changes to Reviewer 1 in red and Reviewer 2 in blue.

**Response to Ramesh Glückler**

**General comments (in red)**

In their manuscript "Fire activity in the northern Arctic tundra now exceeds late Holocene levels, driven by increasing dryness and shrub expansion", the authors explore timely relationships between past wildfire dynamics, vegetation, and hydrology in the Alaskan tundra. For that they use a combination of new and previously published paleoecological data on past wildfire activity, vegetation composition, and the peatland water table, and combine this paleoecological data with more recent fire observations from remote sensing. The main result, summarized in the manuscript's title, is based on increased reconstructed fire activity after 1880 CE, coinciding with drying and shrub encroachment. Overall, the manuscript presents a valuable perspective on past wildfire activity, especially by its approach to compile data from multiple sites and to include a reconstruction of the peatland water table. The comparison of charcoal-based fire reconstructions with more recent fire observations is intriguing and contributes to ongoing efforts of determining regional charcoal source area. The manuscript content fits well within the scope of Biogeosciences, and would be an interesting contribution to the journal for researchers across various disciplines, such as paleoecology, peatland ecology, fire ecology, hydrology, or tundra vegetation dynamics. However, in its current state the manuscript would in my opinion benefit from some further polishing and some clarifications regarding the proposed drivers behind the reconstructed fire activity. Before publication I would therefore recommend some moderate editing, including clarifications regarding the applied methods and interpretations, a more comprehensive description of the results, and some changes to the structure of the discussion to guide the reader more smoothly. My suggestions therefore do not concern the great underlying data and analysis, but rather potential improvements in the way these are presented and discussed, so the manuscript can stand at its full potential. From my perspective, a correspondingly revised version should be very much considered for publication in Biogeosciences. Please find more detailed comments below:

Specific comments: Manuscript structure and clarifications

1. From my perspective, some changes to the manuscript's structure would improve reading flow and potentially clarify some remaining questions. First, it was not always completely clear to me which data were already published before (and exactly where), and which were newly created, and for some of the data, I was missing a clear description. I think all of this may be mostly related to the short results section, which only includes sub-sections for the fire-related data (charcoal and remote sensing). However, in the methods section, also new chronological data, data on vegetation (from pollen and plant macrofossils), and data for the water table (testate amoebae) is mentioned. I think even if previously published, since they were also processed as part of this manuscript, they should also be featured and briefly described in the results section, where they are currently missing. Expanding the manuscript in this regard for methods would be appreciated as well, since it is currently not always

clear which methods were applied due to the short nature of the description (e.g., in L122 a composite of shrub data (and later water table data) is mentioned, but lacking details on how exactly it was composited, and in L126 the abbreviation "DTW" is introduced, but not clearly described). If the testate amoebae data was published before, what were the main outcomes? How would the newly composited water table curve be described? Introducing the data in a bit more detail would help the reader follow along in the discussion.

**R: Methods**

We have expanded the Methods section in the main text (3.1 Chronology) and the Appendices to include the radiocarbon and lead dates (Table B1a,b) and the age depth models for all sites (Appendix B) along with details on chronology for the three unpublished sites, SG, RH, and DL.

We have clarified the calculation of interpolated charcoal accumulation rates to account for changes in sediment accumulation rates (3.2 Macroscopic charcoal inferred fire history, lines 126-129).

We also created separate subchapters to expand the methodology of plant macrofossils and pollen-based vegetation reconstructions (3.3. Plant macrofossils and pollen-based reconstruction of vegetation dynamics, on lines 131-138) and testate amoebae water table DWT reconstruction (3.4 Testate amoebae reconstruction of hydrological changes, on lines 140-144), as well on the creation of their respective composite records (3.5 Composite records of fire, vegetation and hydrological changes on lines 146-160).

**Results**

Following the same logic as for methods, Chapter 4.1 Charcoal-based biomass burning, includes now the description of interpolated CHAR values, along with a slightly expanded description of the trends in CHAR (lines 182-186).

We have also created separate additional subchapters that include brief descriptions of results from pollen and plant macrofossil-based vegetation reconstructions (4.3 Vegetation changes, on lines 207-216), as well as testate amoebae water table DWT reconstructions (4.4 Hydrological changes on lines 218-224).

- 2. Speaking of the discussion, I would personally not start with the methodological interpretation of charcoal source area, but rather with the most important aspects of the manuscript mentioned in the title although this is definitely a subjective preference. However, also here some aspects should be expanded upon. For example, in L195 historical fire observations were mentioned, but not further described. Which kind of fire activity was recorded between 1880 to 1920? Does it fit to the reconstructed charcoal-based signal? Finally, instead of the last sentence of the discussion, I suggest to include an actual conclusion section that briefly summarizes the findings and provides an additional outlook. I feel that would provide a smoother end to the manuscript.
- R: Thank you for the suggestions regarding the manuscript's structure. The reason for starting the Discussion with the methodological Interpretation of charcoal source areas is that readers have an overview of what charcoal records represent in terms of distance from the fires and biomass burnt. Once these aspects are established, readers will be able to understand the long-term charcoal record more effectively. For this reason, we would like to maintain the manuscript's structure, starting with the Interpretation of charcoal source areas. However, we fully agree with the reviewer's suggestion to include a separate conclusion section, in which information on the main findings from this study, along with further outlook, is presented on L. 328-239.

Regarding historical fire observations, they refer to evidence coming from aerial imagery as well as anecdotal records as presented in Miller et al. (2023). "Historical evidence from 1948 aerial imagery further documents tundra fires that burned sometime between 1880 and 1920 on the northern Brooks Range, i.e, near the Ketik and Meade Rivers, Shivugak, and Starfish Bluff, with additional anecdotal records mentioning fires in 1952 and 1959 on the Arctic Slope (Miller et al., 2023)." L 262-264.

Fire Interpretation before the drought? I generally agree with the main conclusion of relating moisture conditions, fuel types, and fire activity, as this would be expected, and it would certainly be great to see this relationship clearly in paleoecological data. However, some aspects of the suggested relationships, as seen in the actual data, may need a bit more explanation. In L209, it is stated that the driest conditions of the Late Holocene (during the past few decades) coincided with the most intense/severe fire period. However, looking at Fig. 3a it seems rather that severe fires occurred between c. 1900 to 1970 CE, whereas in Fig. 3c it seems that drier-than-average conditions only occurred since ca. 2000 CE, when the levels of biomass burned were very low. A similar discrepancy is evident in the timing of increased shrub counts in Fig. 3b when compared to the reconstructed biomass burning, even though in L225 it says that woody biomass would have promoted fires. Somehow it seems to me as though the level of biomass burning increased first, and the shrubs and drying followed – but what would then be the driver behind the fire activity? How can these temporal differences be understood in light of the proposed relationships, and does proxy taphonomy play a role as well? In case I did not misinterpret the plots, I think that these aspects should be included in more detail in the discussion.

R: Following the reviewer's two suggestions, we have calculated the charcoal accumulation rate (CHAR) on charcoal values interpolated at the median temporal resolution of each record to account for marked changes in the sedimentation rate in records in the uppermost sediments compared to the rest of the core. This interpolated composite CHAR record aligns temporally more closely with composite records in DTW and vegetation. However, some temporary asynchrony changes in fire, shrub abundance and moisture persist and likely connected techonomic processes. Please see full further explanations on potential reasons on lines 317-326.

**Technical corrections:**

L34: Suggest to re-phrase sentences, e.g. "[...] charcoal records in combination with data on vegetation, hydrology, and satellite-derived fire observations from [...]". In the following sentence: "A regional composite of charcoal records shows [...]"

R: Done, thought slightly modified: "The composite charcoal record shows minimal fire activity..." L. 36.

L44: Maybe instead of ending a bit abruptly after this result regarding source area, a sentence briefly summarizing the importance of the study, or providing some outlook, would be fitting here?

R: Thank you. The ending abstract now reads: "Our study emphasizes the significance of long-term, multidisciplinary research in documenting moisture—vegetation—fire feedbacks that influence tundra fire regimes. Ultimately, this long-term fire dynamic study provides critical context for evaluating recent changes and incorporating tundra peatland fire risk into global climate mitigation strategies" 1.45-47.

L46: Suggest to re-phrase to e.g. "[...] increase in the number, size, and intensity of individual fires, and in the length and duration of fire seasons [...]"

R: Done: 'Short-term records of fire activity derived from satellite observations indicate that many northern tundra regions have experienced an unprecedented increase in the number, size, and intensity of individual fires, as well as the length of the fire seasons in recent decades (Descals et al., 2020; Scholten et al., 2021, 2024)" L. 50-52.

R: L52: Corrected to "Sayedi" et al., 2024; L. 53.

L53: Suggest to re-phrase here, e.g. "[...] contributed to a deviation from the previously low wildfire activity"

R: Corrected to "Recent rising temperatures, increased lightning frequency, changes in peatland hydrology, and greater shrub biomass contributed to a deviation from the previously low fire activity (Scholten et al., 2024)". L.57-59.

R: L67 Corrected to "peatlands", L. 75

L66, 70: No reference for "Vachula, 2020" or "Vachula et al., 2020" is listed in the reference list. Do you refer to Vachula (2021) in L70 (see references below)? Maybe just double-check that all citations are mentioned in the reference list and vice-versa.

R: We refer to Vachula RS, Sae-Lim J, Russell JM. Sedimentary charcoal proxy records of fire in Alaskan tundra ecosystems. Palaeogeography, Palaeoclimatology, Palaeoecology. 1;541:109564. This title is now incorporated into the reference list.

L73: "tussock sedges-moss-dwarf shrubs" seems like a bit of a clunky term, maybe there is another way to refer to this vegetative zone? Also note additional space before "dwarf"

R: We will simplify the term as follow: "We combined satellite and charcoal records, primarily from the tussock and shrubby tussock tundra zones, with data on local tussock and shrubby tussock tundra zone" L.88-89.

L74: not sure if an "en"-dash is needed in "testate amoebae-based" – maybe double-check journal guidelines. and L148: In this case for ranges I think that "en"-dashes should indeed be used. Maybe consult the journal guidelines for a revision (same in L165, L170)

L85: Annual precipitation averages 25 mm [...]" – is that the average annual precipitation sum across multiple years? For that it seems quite low. Or do you refer to the average across all months of a year? In that case, I think that the first would provide better context. Maybe it is good to just clarify which average this refers to.

R: Thank you for spotting the mistake, a zero was missing! "Annual precipitation averages 250 mm". L 100-101.

L92: I am not sure I can follow the description of how cores were retrieved with a shovel, maybe you can provide a bit more detail?

R: We used a long-blade shovel (50 cm long, 15 cm wide) to dig a hole into the mineral substrate. Then, we cut a peat layer, retrieved it manually, and cut it again with a sharp knife to cleat it and place into PVC tubes fort transportation. Refined text: "Peat cores (monoliths) were retrieved using a long-bladed shovel, dug to the mineral substrate, and the sediments from the hollow were retrieved using a Russian-type corer here were in from of a peat monolith". L. 108-109.

L101: Sites "Ga I and GA II" or sometimes written with a space, sometimes without (e.g., L126)

R: We have uniformised all sites name to version; NO space.

L107: Suggest to include here: "[...] intervals across cores from all sites"

R: Rephrased to:" We reconstructed changes in local-scale fire activity based on macroscopic charcoal analysis of 1–2 cm3 sample volumes taken at 1 cm contiguous intervals across cores from all sites" L.123-124.

L107: Which bleaching agent was used in the procedure? This should be stated in any case, but especially since previous analyses found that certain bleaching agents may dissolve charcoal particles from low-intensity fires (Constantine and Mooney, 2021), which (I suppose) are the kind of fires one may expect at these sites

R: We used sodium hypochlorite (domestic/household bleach): "Sample preparation involved overnight bleaching using sodium hypochlorite (domestic bleach) and wet sieving through a 160 µm mesh". L 118-119.

L112: Sometimes "charcoal influx" is used, sometimes "charcoal accumulation rate (CHAR)". I suggest to standardize these terms across the manuscript, as they seem to be used interchangeably.

R: Thank you, for consistence we used charcoal accumulation rate through the manuscript.

L112: The unit description here is missing some "minus" and numbers – I think it should be "particles  $cm^2 yr^{-1}$ ". This also goes for other instances throughout the manuscript (e.g., right afterwards: "particles  $cm^{-3}$ " and "year  $cm^{-1}$ ", or in L146) – Also note that in some figure captions, the unit is written as " $\#/cm^2/yr$ ", I'd recommend to standardize one type across the manuscript.

R: Thank you, we have uniformised the name and units for charcoal as particles cm-2 yr-1

L121: "smoothed"? R: Corrected to smoothed.

L136: Add space: "70 km" and standardize across the manuscript. R: Corrected L.

L137: "[...] the distance from the nearest [...]"

R: "We calculated the distance from each fire point and fire polygon from the Dalton Highway as well as the distance from the nearest five fire points to each of the fossil record". L177-178.

L145: Add space: "mean = 0.04 [...]". R: Added

L152: Suggest to replace "the" with "an"

R: "The charcoal morphological record indicates an almost equal representation of herbaceous" L 194.

L154: "macro charcoal" – double check that this term is always used in correct and equal spelling across the manuscript (suggest to use either "macro-charcoal", "macrocharcoal", or "macroscopic charcoal")

R: We standardised the term to macrocharcoal through the manuscript.

L158: Suggest to start with "Our satellite [...]" and use the term "buffer" instead of "radius"?

R: Changed to: "Our satellite imagery record of fire points within a 70 km buffer along the Dalton Highway shows that 36 fires occurred between 1969 and 2023". L. 192-195.

L157: The "extremely large fire" is 103.896 ha or 103,896 ha? It may be good to double-check that the use of comma/dot for decimals is always correct and according to the international norm, including in the corresponding figure caption etc. In L160, a space is missing before "2 ha", and later a "=" is missing in "median 115 ha".

R: It was a very large fire of 103896 ha. For clarity we remove the dot

L168: Before, no spaces were used when mentioning e.g. "Fig. 4c". Recommend to standardize across the manuscript. R: Standardised to dot.

L171: "[...] while the satellite data during 2001–2017" is not a full sentence, maybe this part could be re-phrased?

R: Rephrased to: "However, the CHAR record suggests greater burning activity around the 1970s. In contrast, satellite data indicate maximum fire size and frequency between 2001 and 2017 CE (Fig. 4a)." L 205-206.

*L176: Delete space at beginning of the line* R: Done

L180: I would have expected some citations already within this sentence, to make it clear which study is related to the 2 km source area or the wider area. In the next sentence "Vachula et al., 2020" is mentioned, which may again may need to be corrected.

R: Rephrased to accommodate this. Please see L. 226-232.

L186: Missing comma in citation, R: Added: Pereboom et al., 2020. L 246

L191: I am not sure data on fire intensity was presented here? As it's a similar theme, I would also refrain from mentioning an "intense fire period" (L209) and rather use the term "severe", to reduce potential confusion about fire regime terminology.

R: The studies cited, at least those from satellite images above refer to fire frequency and size. L 256-257.

L211: "Smaller-amplitude deepening of the water table [...]"? R: We replaced this with lower amplitude drying. L. 254.

L220: Suggest to add commas: "[...] fuel type and availability, and fire, particularly [...]"

Figure 2: I am not sure if I understand this sentence in the caption: "The total charcoal influx represents the unidentified charcoal." — could that be re-phrased and clarified? Also, note the CHAR unit being written differently here than in the main text. In L432, standardize the use of spaces and capitals when defining "Common Era" (including other instances across the manuscript). In the last sentence of the caption, it says that the smaller plots on the right side cover the last two centuries, yet their x-axes seem only to reach back to 1960 CE?

R: Apologies for the clumpy sentence and inconsistency. Total charcoal influx refers to the total amount of charcoal at each site, as opposed to the charcoal that was differentiated into morphological types. The captions now reads: "Figure 2. Total interpolated charcoal accumulation rates (black bars) and separated on woody charcoal morphologies (green bars; particles cm-2 yr-1) at nine sites along the Dalton Highway. The numbers in brackets indicate the site numbers as marked in Fig. 1. At sites with identified woody morphologies, the difference between the total charcoal and the woody morphology represents the morphologies formed by herbs (grass leaves, stems of forbs, and grasses) and broadleaves. On the right-hand side are trends in charcoal accumulation rates over the last century. The orange rectangle highlights the period of marked increase in total charcoal influx. BCE=before Common Era; CE Common Era", lines 609-614. We aalso revised this figure to accommodate esthetical suggestions.

Figure 3: In caption in L442: Is it correct to write "[...] as counts. (c) Composite record [...]"? Regarding the figure itself, would there be a way to separate the inserted plots more clearly from the main plots, so the numbers don't overlap with the background plot? For example, they could all be decreased in size a little bit and inserted on the left side of the respective main plot, where there is more space. Two of the inserted plots also currently miss x-axes. The y-axis description of plot a) is not completely clear to me — it should probably say that higher values mean more biomass burning, but from just the one arrow/symbol it is not fully clear. Maybe it would be better just to use two arrows and a description, such as in plot c)? Also, it seems that the x-axis line in plot c) has a variable thickness, whereas it is not visible at all in some of the other inserted plots.

R: Figure 3 was revised to accommodate reviewers' suggestion, please see the revised figure on L. 595

Figure 4: In caption, I suggest to list the cut peaks as "a) 2001 (341 ha), 2007 (659 ha); b) 2007 (103.896 ha)" – note the b) 2007 value again here, is it really supposed to be 103.9 ha? In the last sentence, standardize use of spaces around "=". In the figure itself, I am wondering if a) and b) could not be combined into a single plot, with the bars of different width in the back and CHAR in the front, maybe in a different color? Also, is it really "Cumulative charcoal influx" in a)? The compilation method by Blarquez et al. (2014) does not just sum the different records, so it's likely not cumulative – if that was done here, it should be stated in the methods section. In plot c), there are variable distances between the legend entries, and the x-axis line is interrupted.

R: **Figure 4.** a) Comparison of fire frequency and size (ha) represented as fire points (red bars) and fire polygons (orange bars) from satellite imagery with interpolated charcoal influx (black bars; particles cm-2 yr-1): \*Note: The full extent of large fire occurring in 2007 (103896 ha) is not shown to improve visibility of trends in smaller fire events. b) Distance to nearest five fires from each of the sites: I=denote the nearest and V= the furthest from each of the sites. L.639-643

Table 1: "Coordinates" R: Corrected

Appendix C: Suggest writing either just "particles >500  $\mu$ m" or "particles larger than 500  $\mu$ m".R: Corrected to particles >500  $\mu$ m

Data availability: Ideally, the data could be uploaded during the revision so that the final DOIs can be included in the manuscript, which would make finding the data a lot easier in the future. In any case, I'd recommend to fully reference Neotoma, e.g.: "[...] deposited to the Neotoma Paleoecology Database (www.neotomadb.org; Williams et al., 2018)"

R: We are making the effort of deposit all our datasets to Neotoma. In fact, most of the datasets have already been submitted and perhaps will receive a doi by the time of final publication of this paper.

Supplement S1: "Satellite-based". R: corrected

Author contribution: "conceived the study" R: corrected 1. 287

Affiliations: Suggest to double-check spelling of institutions, order of institutional levels, and the inclusion of the full address

R: WE uniformised this, thank you,

**References mentioned in this review:**

Blarquez, O., Vannière, B., Marlon, J. R., Daniau, A.-L., Power, M. J., Brewer, S., & Bartlein, P. J. (2014). paleofire: An R package to analyse sedimentary charcoal records from the Global Charcoal Database to reconstruct past biomass burning. Computers & Geosciences, 72, 255–261. https://doi.org/10.1016/j.cageo.2014.07.020

Constantine, M., & Mooney, S. (2021). Widely used charcoal analysis method in paleo studies involving NaOCl results in loss of charcoal formed below 400°C. The Holocene, 09596836211041740. https://doi.org/10.1177/09596836211041740

Vachula, R. S. (2021). A meta-analytical approach to understanding the charcoal source area problem. Palaeogeography, Palaeoclimatology, Palaeoecology, 562, 110111. https://doi.org/10.1016/j.palaeo.2020.110111

Williams, J. W., Grimm, E. C., Blois, J. L., Charles, D. F., Davis, E. B., Goring, S. J., Graham, R. W., Smith, A. J., Anderson, M., Arroyo-Cabrales, J., Ashworth, A. C., Betancourt, J. L., Bills, B. W., Booth, R. K., Buckland, P. I., Curry, B. B., Giesecke, T., Jackson, S. T., Latorre, C., ... Takahara, H. (2018). The Neotoma Paleoecology Database, a multiproxy, international, community-curated data resource. Quaternary Research, 89(1), 156–177. https://doi.org/10.1017/qua.2017.105

**Response to Review 2 (blue)**

**General comments**

This manuscript contributes to our understanding of fire-regime variability on millennial to centennial timescales on the Alaskan North Slope along the Dalton Highway corridor. The authors use macroscopic charcoal counts from peatland soils to reconstruct charcoal influx associated with past fires. They also reconstruct vegetation compositional changes at local and regional scales using a combination of macrofossil and pollen identification from their sites, respectively. These data, along with previous reconstructions for water table changes from testate amoebae at a subset of sites provides information for qualitatively evaluating feedbacks between hydrology, vegetation, and landscape flammability. The authors also use spatially explicit fire datasets to constrain the source area of the macroscopic charcoal counts. Overall, this is a well-developed dataset and robust contribution that expands our understanding of the palaeoecological history of this rapidly changing region. I appreciate that the findings are generally not overstated, with the exception of the manuscript title that makes a very bold claim which may only tenuously be supported by the data (please see specific comments). Overall, I think that some revisions are needed to clarify the interpretations of the datasets and the overarching conclusions.

Specifically, greater attention to chronological control and constraints, as well as a more nuanced explanation of the some of the temporal patterns, would improve the manuscript. I hope the specific comments I provide below are helpful to the authors.

**Specific comments:**

**Specific comments:**

- 1. I would really like to see some background information in the introduction as to the sources of charcoal in peatlands and the the pros/cons of using that setting for fire history reconstructions. This does not have to be overly long, but a bit of general background on the methods of extracting fire reconstructions from peatlands specifically would be quite useful to the reader.
- **R:** Our background information's now reads: "Compared to lakes, peatlands provide local-scale reconstructions of past fire regimes, with the source area of charcoal from a fire typically extending only a few kilometres (Conedera et al., 2009; Remy et al., 2018). This fine-scale resolution reflects the pronounced heterogeneity in local peat moisture, vegetation composition and structure, and the relatively small catchment area (Magnan et al., 2012; Remy et al., 2018; Barhoumi et al., 2019; Stivrins et al., 2019; Feurdean et al., 2020; Kuosmanen et al., 2023). Because wildfire patterns are highly variable at a small spatial scale, multi-site peatland reconstructions are necessary to identify broader trends in fire regime dynamics beyond the local scale. L. 76-82.
- 2. Chronological control is probably the most important part of any paleo study. I would highly recommend adding the age-depth models for all nine sites to the appendix so that readers can easily examine the fidelity of all models. A master table with all AMS and 210Pb ages per depth for all sites would also be useful to add the appendix.
- R: Thank you for this observation. Following the reviewer' 1 suggestions, in the revised version of the paper, we have provided an extended Table. B1a,b including the AMS14C and Pb210 dates, as well as age depth models for all sites.
- 3.One of the major claims in this paper is that fire activity since 1880 CE was higher than the previous millennia. At first glance, the data presented in Figure 2 is compelling (and exciting!). However, it is unclear to me if the charcoal influx data for each site has been interpolated to the median or average sampling resolution for each record. The reason I am concerned about this is that without a consistent

temporal resolution, trends in influx could solely be a function of changes in the sedimentation rate. Specifically, the sedimentation rate changes dramatically in all records presented in the appendix in the uppermost sediments compared to the rest of the core. Equal time sampling will be important to compare past to recent trends, and should be explained carefully in the text if the take home is indeed that fire regimes have changed.

R: Thank you for this observation regarding the effect of changing sediment accumulation rates on trends in charcoal accumulation rates (CHAR). We have subsampled all records continuously at 1 cm resolution. In the original version of the paper, the charcoal accumulation rates have not been interpolated to median or average sampling resolution before the construction of the composite CHAR record (Fig 3), nor the CHAR plots at individual sites (Fig. 2). In the revision version we have interpolated the charcoal data to a median resolution for each record and used the interpolated charcoal accumulation rates in the composite record (n = 9)" L. 126-129, 146-149.

The trends in interpolated individual and composite charcoal records are slightly different than untransformed data. However, it preserves the increasing charcoal accumulation trend in recent times, implying that the key takeaway message — that recent fire activity is the highest compared to the late Holocene level — remains valid. Please see ". L. 181-186, Figs 2,3.

3. Another point in regard to chronological control is line 37-38 in the abstract. Is it possible that the heterogenous fire patterns observed in the past compared to more recent sediments is a function of both 1) lower temporal resolution in the deeper parts of the peat cores and 2) the much larger age error of the deeper sediments? I think these types of issues should be discussed explicitly in the text to provide a more conservative and nuanced interpretation of the data. And the comments from the editor; AUTHOR RESPONSE: Even though the temporal trends in CHAR are slightly shifted compared to the original, untransformed record, this interpolation does not affect the original observed pattern; that fire patterns were more heterogeneous among sites in the past compared to more recent times." Here the reviewer asked for greater discussion on issues surrounding lower sampling resolution in older core section and larger errors in dating these sections, but the Author Response does not respond directly to this request. This is a very valid comment and will require careful analysis and examination in the resubmitted manuscript. I ask the Authors to carefully examine the reviewer requests and respond all of the comments. to parts

R: We have extended slightly the interpretation of heterogenous vs homogenous pattern and the influence of age control. While a lower temporal resolution may have affected the spatial reconstruction of the occurrence of fire activity at multiple sites, past periods of enhanced fire activity at multiple sites were also reconstructed. "Another finding of our study is that individual charcoal records show a heterogeneous pattern in fire occurrence before 1950 CE, and a more homogeneous one thereafter, consistent with the emergence of more regularly occurring fires in recent decades. Lower temporal resolution and larger age control errors in the older part of the sediments could have also contributed to this apparent heterogeneity. However, the more elevated values in biomass burning around 1200 and 1600 CE, visible at multiple sites, suggests that homogenous fire occurrence across sites in the distant past could also be reconstructed" L. 273-279.

4.In terms of the analysis comparing the timing of recent fires and the distance of fires from the depositional sites (summarized in figure 4) --- I am not sure I understand this analysis and it would be helpful to see some clarification. From my understanding, peaks in the charcoal influx data seem to capture deposition of charcoal from long distances, and yet do not capture the larger fires (specifically the Anaktuvuk River Fire that burned ~1000 km2)? I think the disconnect here is that I do not quite

understand how the timing of the fire peaks were determined, and/or what the age model error around those timings actually is. From what I can tell, peak analysis to identify which peaks are most likely to represent fire events versus redeposition processes were not done on these cores. These types of peak analyses that are commonly used in lake sediment studies (with minimum count screening, signal-tonoise analyses, etc) may not be appropriate for peat cores (see comment #1), which is fine. However, I think it is important to carefully clarify how these influx peaks can capture fires from long distances only at certain time periods. I think the text from lines 180-187 is not substantial enough to really explain this disconnect. What I would recommend is to 1) carefully explain how the timing of fire peaks was determined in the paleorecords and the temporal span of each peak, 2) clearly explain how many peaks seem to correspond to a nearby fires and how many do not, and then 3) where they do or do not, explain the processes that may explain the pattern (i.e., are peaks more related to what actually burned in the fire, or the distance from the fires to the site, or both). I think articulating this carefully in this way will greatly help the manuscript. In lines 184-187 – the authors mention that gramminoids have lower charcoal rendition rate per unit biomass because they may be completely consumed in fires. Would this potential preservation bias based on vegetation type partially explain why more charcoal seems to be present in the records when shrubs are more common the landscape? I think this in an important point to address in the interpretation. While it makes intuitive sense that more flammable species and higher biomass on the landscape would result in larger and/or more frequent burning (and/or create positive feedbacks), preservation biases from graminoid burning versus shrub burning could also be important to address.

R: Regarding data type: In figure 4, the peak in charcoal refers to elevated values in the charcoal accumulation rate in several consecutive samples. We have not examined charcoal peaks as in Higuera et al. (2009), which implies statistical separation of charcoal peaks from charcoal background, peak evaluation (i.e., minimum count screening, signal-to-noise analyses, etc.). The main reason is the very low charcoal counts and numerous zero samples, and at some site low median sedimentation rate relative to the entire record, which argue against using CHAR analysis (Higuera et al., 2009). Our preliminary peak analysis test using CHAR analysis, confirmed that charcoal peaks could not be satisfactory separated from background at many sites.

R: Regarding the correlations between the charcoal peaks and fire sources, Figure 4 and Table S1 show that most documented fires occurred at considerable distances (3–29 km) from our sampling sites, and the fire source areas at our sites overlapped. Although the reviewer's suggestions provide an elegant way to address this issue, it remains challenging to determine whether the elevated CHAR values primarily reflect fire size, fire proximity, or a combination of both. Nevertheless, in the revised version of the manuscript, we have expanded on the potential connection between charcoal values and the site distance from the fires, the size of the fire, and the type of biomass burned. Please see the revised text on lines 235-255.

3. Modern fire data - please clarify in the methods how the area burned for the point data was estimated. Was this based on field surveys by the AICC? For the polygon data, was estimated area burned based soley on the size of the polygon, or were non-vegetated surfaces removed from the area burned estimate?

R: In case of fire polygons, the area burned was based solely on the size of the polygon; non-vegetated surfaces were not removed. In case of fire point data, the reported incident size was estimated from field observations or calculated using GIS from perimeter polygons with no possibility to trace back how

this was established (Miller et al., 2023). Therefore, the recovered size of fire from fire point data should be treated with caution. L. 167-168.

4. The final comment I have is in regards to the water table analyses. While I think it is awesome to have that type of information in the study (especially considering how important this could be to future fire regime changes), it was unclear to me how spatially consistent water table changes actually are in this region. For example, I am thinking about how that area has large ice wedges that can cause subsidence and intersection with the water table as they thaw, and how fires themselves may cause surface subsidence through these thaw processes. Thus...how sure are the authors that water table changes from these reconstructions actually apply to the whole area and/or that these records are appropriate to combine into a composite? I think a bit more text explaining the logic here would go a long way to improving the interpretation.

R: Our study area does not lie in an ice wedges region. In all instances, we cored from the hummock area of each peatland. No fires have been documented directly at the coring sites between 1969 and the present based on satellite images. Therefore, it is unlikely that fire has caused the thaw of permafrost and local surface subsidence.

**Technical corrections:**

- 1. Figure 1 and Table 1: Please add the site codes to the map OR the site numbers to the table. It is not possible to easily figure out which site is which on the map.
  - R: The sites codes are now added to the revised Table 1.
- 2. Table 1: I would recommend adding original citations for previously cored sites to the data table. Also, it is not clear to me if the charcoal, pollen, and macrofossil analyses are new for all sites, or only for a subset of sites. Clarification of new versus pre-existing analyses in the methods section would be very helpful.

R: We have added the original reference to each core. In the main text we have also made it clearer which datasets are new and which published. Please see the newly added text marked in red and blue in the Methods.

Figure 1: I can only see two 2007 fires in the polygon data on this map. I am assuming that the other 7 fires occurring in 1983, 1990, 2001, 2009, and 2017 fires are so small that they are obscured by the point data? In any case, I think the solution here would be to use a single color for the fire polygons and label the years of these polygons directly on the map so that the reader can see them (or some other workable solution that allows the fire polygons to be seen by the reader).

R: Thank you. To improve the readability of Figure 1, we have split it into two separate Panels, one for the fire polygon (Figure 1a) and the other for the fire points data (Figure 1 b).

*Line 53* – "low fire patterns" is a confusing phrase. Please reword and clarify the meaning.

R: Thank you for pointing it out. We have revised to: Individual charcoal records show a spatially heterogeneous pattern in fire occurrence before 1950 CE, and a more homogeneous one thereafter. L. 38-39.

- 3. Line 63 feedback should be plural R: Corrected. L.67
- 4. Figure 2 caption parentheses bracket missing in the first line R: Corrected

Your sincerely,

Angelica Feurdean on the behalf of all coauthors

---

## Author Response (AR2)

We thank the reviewers and the editor for providing feedback and valuable comments on the manuscript. We have incorporated final the technical remarks from the reviewer 1.

**Remarks from the preceding review file validation**

Figure B1 has separate captions for each panel. This is not allowed according to our policy. Please combine all panels in one single figure using (a), (b), (c) labels, etc. and combine all captions in one single caption.

R: Figure B1 one has been changed to show one caption. However the panels are large, therefore I cannot combined them in one single figure. I therefore provided 5 separate figures, including the name of each panel on each figure.

**Editor Public justification**

I would like to thank both the authors and reviewers for their time and commitment to moving this manuscript to production and I am now happy to accept it for publication subject to the minor, technical corrections suggested by reviewer 1. Please incorporate those into the final version of the manuscript for processing. Congratulations to the authors.

Best regards,

**Response to Ramesh Glückler**

From my perspective, the authors well implemented most suggestions from the previous reviews. The results and methods sections now feature a brief description of all analyses that are either newly done or used from previous publications, and a separate conclusion section improved the manuscript structure. The authors provide good reasoning for starting the discussion section with some methodological paragraphs. In general, many parts of the initial manuscript were improved, ambiguity reduced, mistakes corrected, and clarifications added where it was needed. Importantly, the potential temporal mismatch behind the main interpretation is now addressed more clearly in the discussion. I also appreciate the revised figures and the submission of the data to a publicly accessible database. As in the initial review, I think that this manuscript is worthy of publication in Biogeosciences. After reading the revised manuscript, I was left with the impression that some minor polishing would still be beneficial, but this concerns mostly small technical or formatting-related issues. I therefore recommend the acceptance of the manuscript after some additional technical corrections by the authors. considering also the below. remaining suggestions

Other remarks:

L35: correct "from in"

R: ... "with satellite-derived fire datasets from northern Arctic". L.35

L36: Is there a double space between sentences? (also potentially in L239, 291, 305, 686) R: We have removed double space from several places in the manuscript.

L44: "...the charcoal source area of our tundra fire..." – suggest rephrasing this, e.g., "our tundra

fire reconstruction", and also add plural for "kilometers"

R: ..."reconstruction encompasses broader landscapes over tens of kilometres." L. 44

L71: I still cannot find a reference for "Vachula, 2020", which is also mentioned in L257 and L269. Note also that in L629 the new reference to Vachula et al. is missing the publication year.

R: Vachula RS, Sae-Lim J, Russell JM. Sedimentary charcoal proxy records of fire in Alaskan tundra ecosystems. Palaeogeography, Palaeoclimatology, Palaeoecology 2020, 1, 541, 109564. https://doi.org/10.1016/j.palaeo.2019.109564 L. 625

L73: Citation should be "Frank..." (also in L270, 272, 292)

R: Corrected to Frank-DePue and Chipman, 2025 here and the other lines.

L126: minus missing in concentration unit, same in L179, L180

R: Minus added to units, see L. 126 178, 179

L132: Double-check use of brackets here, it was slightly confusing to read and not sure if every opening bracket has a corresponding closing one. Maybe present rather as done in section 3.4? R: adjusted to "We extracted shrub abundance from existing plant macrofossil and pollen records from TFSI, TFSII (Gałka et al., 2018), GaI, GaII (Gałka et al., 2023), NBb, and Erin (Gałka et al., 2025). L.131-132

L150: Maybe specify the baseline period range more clearly – does 2 ka refer to the 2000 years BP, i.e., before 1950CE?

R: "... standardization using a baseline period of last 2000 years." L.150

L157: Standardize use of spaces around "=" (see also e.g. L169)

R: Space removed

L158: Here DTW is used as an abbreviation for "depth-to-water table", before it was used for "water table depth" – suggest introducing the abbreviation only once and for the correct corresponding term

R: We standardized the term to: "Changes in peatland hydrology were assessed using existing testate amoeba depth-to-water level (DTW) ..." L. 141, 156, 157.

L240: I would suggest standardizing the use of en-dashes for ranges across the manuscript, as often a normal minus sign is used instead

R: Standardised to minus sign.

L301: I think there's an opening bracket missing R:" ...(Fig. 3)." L.300.

L303: "this increase"?

R: "...between this increase in fire activity and expanded shrub cover ..." L 302

Fig. 3: For the topmost plot, the description "High" is slightly cut off, making the "g" difficult to read – maybe this could be easily adjusted.

R: corected too new Fig.3.

Kind regards, Angelica Feurdean on behalf of the coauthors